# Biased genome editing using the local accumulation of DSB repair molecules system

Shota Nakade[1], Keiji Mochida[1], Atsushi Kunii[1], Kazuki Nakamae[1], Tomomi Aida[2,3], Kohichi Tanaka[2], Naoaki Sakamoto[1], Tetsushi Sakuma [1] & Takashi Yamamoto[1]

Selective genome editing such as gene knock-in has recently been achieved by administration of chemical enhancer or inhibitor of particular DNA double-strand break (DSB) repair pathways, as well as overexpression of pathway-specific genes. In this study, we attempt to enhance the efficiency further to secure robust gene knock-ins, by using the local accumulation of DSB repair molecules (LoAD) system. We identify CtIP as a strong enhancer of microhomology-mediated end-joining (MMEJ) repair by genetic screening, and show the knock-in-enhancing effect of CtIP LoADing. Next-generation sequencing reveals that CtIP LoADing highly increases the frequency of MMEJ-mediated integration. Selection-free, simultaneous triple gene knock-ins are also achieved with the CtIP-LoADing strategy. Moreover, by replacing the LoADing molecules and targeting strategies, this system can be applied for other specific genome engineering purposes, such as introducing longer deletions for gene disruption, independently introducing multiple mutations without chromosomal deletion, and efficiently incorporating a single-stranded oligodeoxynucleotide donor.

[1] Department of Mathematical and Life Sciences, Graduate School of Science, Hiroshima University, 1-3-1 Kagamiyama, Higashi-Hiroshima, Hiroshima 739-8526, Japan. [2] Laboratory of Molecular Neuroscience, Medical Research Institute (MRI), Tokyo Medical and Dental University (TMDU), 1-5-45 Yushima, Bunkyo-ku, Tokyo 113-8510, Japan. [3] Present address: McGovern Institute for Brain Research, Massachusetts Institute of Technology, 46-3143A, 43 Vassar Street, Cambridge, MA 02139, USA. Correspondence and requests for materials should be addressed to T.S. (email: tetsushi-sakuma@hiroshima-u.ac.jp) or to T.Y. (email: tybig@hiroshima-u.ac.jp)

Recent advances in genome editing technology have enabled functional genomics study in any cell or organism, by disrupting endogenous genes, incorporating polymorphisms or short tags, and integrating exogenous genes, such as fluorescent reporters. Of these techniques, gene cassette knock-in is especially useful for monitoring endogenous gene expression and localization, which is essential for functional annotation of endogenous genes. However, previous studies never achieved simultaneous generation of various patterns or combinations of multiple reporters in multiple genomic loci; although, the capacity of double knock-in was confirmed in mouse androgenetic haploid embryonic stem cells[1]; thus, gene insertion has been independently conducted at each locus, and serial cloning steps have been required for double or triple knock-in at different gene loci.

Gene-labeling strategies assisted by genome editing tools such as CRISPR-Cas9 have typically been mediated by homologous recombination (HR)[2,3], and alternative methods utilizing non-homologous end-joining (NHEJ)[4,5] or microhomology-mediated end-joining (MMEJ)[6–9] have recently been established and refined for highly efficient knock-in. The HR-mediated method often results in highly accurate knock-in, but the throughput of targeting vector construction and integration efficiency are inferior to those of end-joining pathway-mediated strategies. The NHEJ-mediated homology-independent targeted integration (HITI) system is especially useful for in vivo gene knock-in with high efficiency[5]; however, this system cannot assign multiple donors with different genomic loci simultaneously because there are no homology arms on the targeting donor vector for the HITI system. To overcome these shortcomings, the MMEJ-mediated precise integration into target chromosome (PITCh) system[6,8] is expected to facilitate high-throughput, simultaneous generation of multiplex knock-in cell libraries because it utilizes short but distinguishable microhomologies (≤40 bp) and results in superior knock-in efficiency compared with the HR-mediated method.

Here we establish the biased genome editing system applicable in PITCh knock-in and beyond, and show its usefulness by enhancing various intended genome editing outcomes including selection-free generation of triple gene knock-in clones by the LoADed PITCh system.

## Results

**Systems development of biased genome editing.** Although, we previously confirmed that the overexpression of exonuclease 1 (Exo1) enhanced the efficiency of PITCh knock-in[9], we attempted to enhance the efficiency further to secure robust gene knock-ins at various genomic loci. We newly identified CtIP as a strong MMEJ enhancer by genetic screening (Supplementary Fig. 1). CtIP, also known as RB binding protein 8 (RBBP8), is an end-resection enzyme of DNA double-strand break (DSB) ends[10]; thus, its overexpression might induce enhanced generation of single-stranded DNA ends, which is essential for the MMEJ and HR repair pathways[11]. Furthermore, we added MS2 sequences at the stem loops of single-guide RNA (sgRNA), and fused MS2 coat protein with CtIP, repurposing the system previously established for highly active transcriptional control, named synergistic activation mediator (SAM)[12]. Using this local accumulation of DSB repair molecules (LoAD) system, strong pathway choice at the CRISPR target site should be achieved (Fig. 1a). To confirm the effect of MS2-CtIP LoADing, we first designed PITCh-mediated mNeonGreen[13] knock-in in front of the stop codon of the endogenous calnexin (CANX) gene, expressing CANX-mNeonGreen fusion protein (Fig. 1b). FACS analysis revealed that the knock-in-enhancing effect of MS2-CtIP LoADing is much greater than those of CtIP, MS2-Exo1, and

Exo1 (Fig. 1c, Supplementary Fig. 2). The LoADing of other related proteins such as Lig3 and Nbs1 did not significantly enhance the PITCh knock-in (Supplementary Fig. 3). The effect of MS2-CtIP LoADing was also observed at four other gene loci (ATP5B, PARP1, FBL, and RPL11) with about twofold activation on average (Fig. 1d, Supplementary Fig. 4). Expected patterns of fluorescence localization were also confirmed by microscopic observation (Fig. 1e, Supplementary Fig. 5). The knock-in-enhancing effect of MS2-CtIP LoADing was also confirmed in other cell lines, CHO-K1 (Supplementary Fig. 6a) and HeLa (Supplementary Fig. 6b) cells. These assays demonstrated the robust, locus-, and cell type-independent knock-in-enhancing effect of MS2-CtIP LoADing.

We next investigated whether the knock-in cells with the LoADed PITCh system were heterozygous or homozygous. Eight and three single-cell clones showing fluorescence in which mNeonGreen cDNA was knocked-in at the ATP5B locus were established with the LoADed PITCh and conventional PITCh strategies, respectively. Subsequently, knock-in and non-knock-in alleles were simultaneously amplified by out–out PCR analysis (Supplementary Fig. 7a). The results indicated that all three clones obtained with the conventional PITCh system were heterozygous, whereas in two out of eight clones obtained with the LoADed PITCh system, no non-knock-in amplicons were observed, although one of these clones showed another longer amplicon, possibly carrying the plasmid backbone as well as the intended knock-in insert (Supplementary Fig. 7b). These results suggest that homozygous knock-in cells would be established using the LoADed PITCh system even in aneuploid cells, such as HEK293T. Using the same cell clones, we further performed Southern blot analysis to investigate whether off-target integrants were detected (Supplementary Fig. 8a). No evidence of off-target integrants was detected among eight clones, except that one minor band showing slightly longer size than expected was observed in one out of eight clones (Supplementary Fig. 8b). Based on this, high risk of off-target integration was not observed in the LoADed PITCh system, as well as the conventional PITCh system.

Another concern for the LoAD system is cytotoxicity of MS2-CtIP. We co-transfected mCherry-expressing plasmid either with mock, CtIP, MS2-CtIP, or zinc-finger nuclease (ZFN) vector that was previously characterized to have cytotoxicity[9]. FACS analysis revealed that CtIP and MS2-CtIP vectors did not affect cell viability, while the ZFN vector decreased cell viability (Supplementary Fig. 9a). We further analyzed the ability of DSB repair when MS2-CtIP was overexpressed, by performing immunostaining of γ-H2AX staining. The cells transfected with mCherry-expressing vector either with mock-, CtIP-, or MS2-CtIP-vector were administrated with 1 μM etoposide (DSB inducer), and then stained with anti-γ-H2AX antibody (Supplementary Fig. 9b). Imaging analysis revealed that etoposide-induced DSB repair signal (i.e., the percentage of γ-H2AX-positive cells) in MS2-CtIP-transfected cells was greater than that in mock-transfected cells (Supplementary Fig. 9c), suggesting that the overexpression of MS2-CtIP resulted in the activation of DSB repair pathway. Collectively, we observed no cytotoxic effect of MS2-CtIP; rather, it activated the DSB repair pathway.

Furthermore, we created various CtIP mutants previously characterized, and analyzed their functionality in the LoAD system (Supplementary Fig. 10). Charpentier and colleagues created and characterized C-terminally truncated mutant of CtIP (1–296 aa), named homology-dependent repair (HDR) enhancer (HE), its phosphorylation-mimicked mutant named HE (3E), containing S233E/T245E/S276E mutations, and its phosphorylation-blocked mutant named HE(3A), containing S233A/T245A/S276A mutations[14]. In addition, the phosphorylation-mimicked mutations of

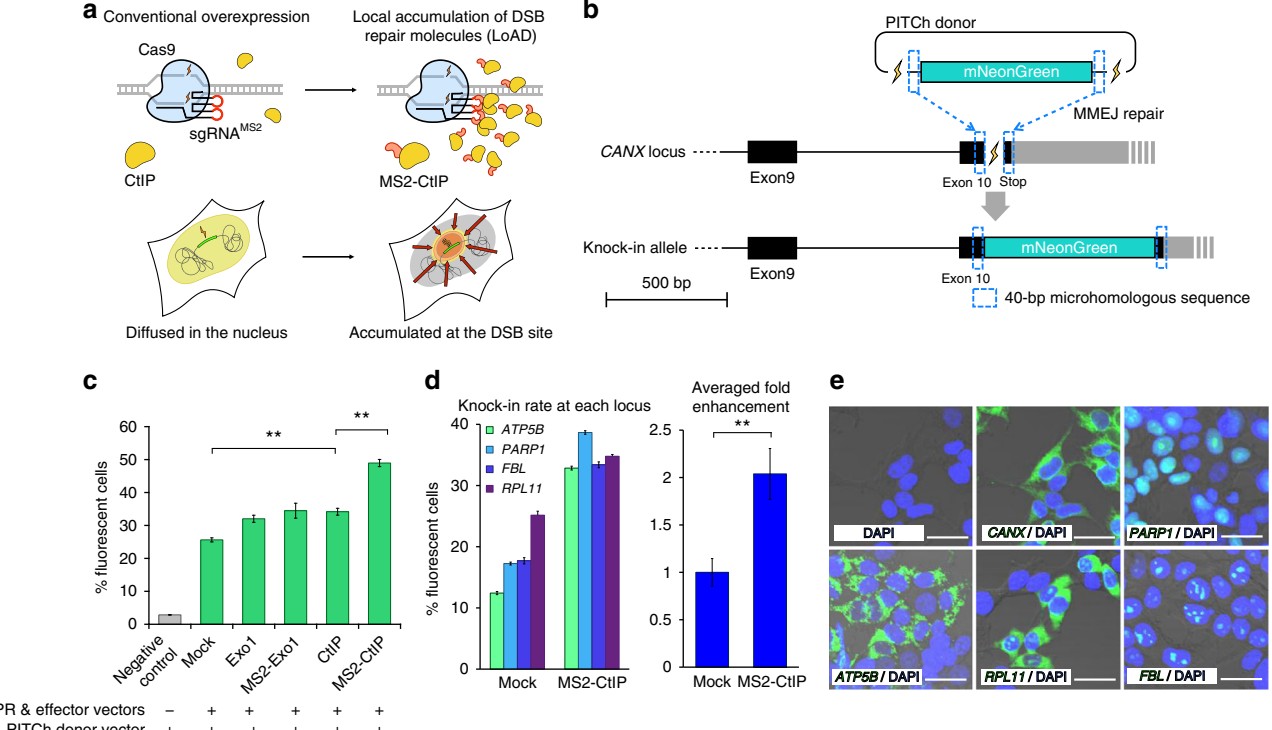

**Fig. 1** Enhancement of knock-in efficiency with the LoAD system. **a** Schematic of conventional overexpression and the LoAD system. Effector proteins such as CtIP can accumulate at the DSB site via the interaction between MS2 RNA loop and MS2 coat protein, resulting in programmable genome editing. sgRNA[MS2], sgRNA with two MS2 RNA loops. **b** Schematic of PITCh-mediated knock-in of fluorescent protein gene, harnessing MMEJ for the predefined, seamless gene cassette integration. This schematic only shows gene targeting in the *CANX* locus, but other gene loci such as *ATB5B*, *PARP1*, *FBL*, and *RPL11* were targeted in a similar way. **c** Knock-in frequency at the *CANX* locus measured by FACS analysis. The recruitment of MS2-CtIP with the LoAD system resulted in about 50% gene knock-in without any selection. Data are expressed as means ± s.e.m. ($n = 3$). **P < 0.01 (Student's *t*-test). **d** The effect of MS-CtIP LoADing was not locus-dependent. The average rate of fold activation was about 1.8-fold. Data are expressed as means ± s.e.m. ($n = 3$). **P < 0.01 (Student's *t*-test). **e** The intended fluorescence localization was observed at all gene loci (CANX, endoplasmic reticulum; PARP1, nucleus; ATP5B, mitochondrion; RPL11, nucleolus, and cytoplasm; FBL, nucleolus). Bar, 30 μm

S327 and T847 (S327E/T847E) reportedly activated the end-resection activity during G1 phase[15,16], and the non-acetylatable mutations of K423, K526, and K604 (3KR) circumvented the requirement of SIRT6 for resection[17]. Based on these findings, we constructed the MS2-CtIP vectors harboring the corresponding mutations (Supplementary Fig. 10a) and analyzed the knock-in enhancing effect with LoADing (Supplementary Fig. 10b). However, the results showed that none of them further enhanced the effect of MS2-CtIP LoADing.

**Characterization of DSB repair pathway choice**. Next, we analyzed the effect of CtIP overexpression and MS2-CtIP LoADing in terms of the accuracy and repair pathway choice. The short 3× Flag tag was knocked-in in this assay to enable sequence analysis covering the full length (left microhomology, knock-in fragment, and right microhomology) by next-generation sequencing (NGS) (Fig. 2a). For the unbiased analysis of all analyzable alleles, including precise knock-in, imprecise knock-in, and non-knock-in ones, we performed out–out PCR of genomic integration sites at the *ATP5B*, *PARP1*, and *RPL11* loci, followed by NGS analysis. Consistent with the FACS analysis, the highest frequency of precise knock-in was observed in MS2-CtIP LoADed samples, followed by CtIP-overexpressing and mock ones, at all three loci (Fig. 2b). In contrast, the frequency of the alleles with insertion and/or deletion (indel) was highest in mock samples, followed by CtIP and MS2-CtIP ones. These results suggested that the LoADed MS2-CtIP selectively enhanced the MMEJ pathway while preventing the error-prone NHEJ. To further characterize the mechanisms of donor

integration, we sorted the read counts by the length of inserted fragments. The results indicated that the frequency of MMEJ-mediated integration was much higher than that of NHEJ-mediated integration, and this tendency was particularly noticeable in MS2-CtIP LoADed samples, followed by CtIP-overexpressing and mock samples (Fig. 2c, Supplementary Fig. 11). Similarly, the frequency of precise knock-in mediated by MMEJ was much higher than that of precise knock-in mediated by NHEJ, and the effect of MS2-CtIP LoADing was also confirmed (Fig. 2d, Supplementary Fig. 12). Our finding that suggested superior efficiency of MMEJ compared with NHEJ is somewhat contrary to the previous observation[5]; however, our data clearly showed that the appropriately structured PITCh donor vector could be incorporated via MMEJ with much higher frequency than via NHEJ. We also analyzed the frequency of base substitutions in and around microhomologous regions utilized for MMEJ (Supplementary Fig. 13). The base substitution properties were similar among the samples of mock, CtIP, and MS2-CtIP; thus, MS2-CtIP LoADing did not decrease the knock-in accuracy. Interestingly, different frequencies and positional patterns of substitutions were observed at each microhomologous sequence. This might have been affected by base composition or other factors such as CpG methylation status and chromatin accessibility. Further data and analysis are needed to reveal the mechanism involved in this. Detailed NGS data are presented in Supplementary Data 2 and 3.

**Parallel generation of multiplex knock-in cell collections**. To efficiently perform simultaneous multiple gene knock-ins, we constructed an integrated multiplex donor vector that produces

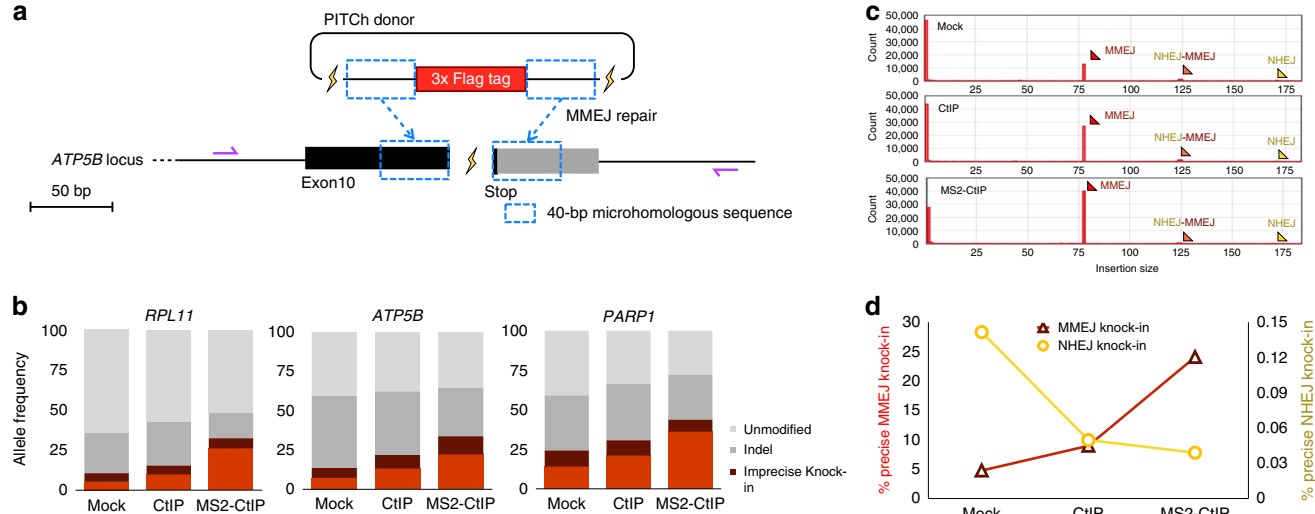

**Fig. 2** High dependence on MMEJ in the enhanced PITCh knock-in. **a** Schematic of PITCh knock-in of a short tag for NGS analysis. An ~50-bp 3× Flag tag sequence was knocked-in, and then all the amplified products using the primers illustrated by purple arrows were analyzed. This schematic only shows gene targeting in the *ATP5B* locus, but other gene loci such as *PARP1* and *RPL11* were targeted in a similar way. **b** Summary of the frequency of the alleles detected by NGS analysis. Precise knock-in includes MMEJ-dependent knock-in alleles without mutations. For the details of data analysis, see Methods available in the online version of this paper. **c** Histograms of the read counts at the *ATP5B* locus, separated by the lengths of inserts. MMEJ, MMEJ/NHEJ, and NHEJ show the expected sizes of the inserts mediated by MMEJ at both knock-in junctions, NHEJ at one junction and MMEJ at the other, and NHEJ at both junctions, respectively, as illustrated in Supplementary Figure 11a. Similar results were obtained at the other two loci (Supplementary Fig. 11c). **d** Percentages of precise knock-in alleles mediated by MMEJ and NHEJ among total NGS reads at the *RPL11* locus. Similar results were obtained at the other two loci (Supplementary Fig. 12b)

three kinds of PITCh donor fragments carrying mNeonGreen, mKate2[18], and TagBFP[19], targeting three different gene loci (Fig. 3a, Supplementary Fig. 14a). We also constructed an all-in-one CRISPR-Cas9 vector based on the previously reported system[20] (Supplementary Fig. 14b). At 72 h after transfection of the integrated donor and all-in-one CRISPR vector with or without MS2-CtIP LoADing vector, we observed a highly heterologous pattern of three kinds of fluorescence without any drug or fluorescence selection (Fig. 3b). The enhancer effect of MS2-CtIP was also observed (Supplementary Fig. 15). Convinced by this observation, we constructed and transfected six types of triple knock-in donors and CRISPR vectors along with the MS2-CtIP vector, and the successful generation of triple knock-in cells in a selection-free manner was suggested by fluorescence observation in all cases (Fig. 3c). To investigate whether the simultaneous generation of various types of knock-in cell clones can be achieved with this strategy, we performed selection-free single-cell cloning of the cells transfected with two types of vector sets for the triple knock-in, resulting in the production of CANX-mNeonGreen, PARP1-mKate2, and ATP5B-TagBFP, or FBL-mNeonGreen, PARP1-mKate2, and ATP5B-TagBFP. Following single transfection and single 96-well cloning for each type of knock-in, various patterns of single, double, and triple fluorescence were observed, suggesting that they were a variety of single, double, and triple knock-in cell clones (Fig. 3d). Among 75 and 93 clones isolated, the numbers and percentages of single, double, and triple knock-in were 13 (17.3%) and 8 (8.6%), 4 (5.3%) and 8 (8.6%), and 1 (1.3%) and 1 (1.1%), respectively (Fig. 3e). Sequencing analysis of 5′ and 3′ junctions of triple knock-in clones revealed that all 12 junctions were precisely joined with MMEJ, except that one junction contained a mutagenic allele along with the correct allele (Fig. 3f). In addition, analysis of various single and double knock-in clones confirmed that 26 out of 30 junctions had the precise knock-in sequence joined by MMEJ (Supplementary Fig. 16). On the other hand, we observed

no visible evidence of unexpected translocations, with the exception of faint bands detected by only one primer set (Supplementary Fig. 17). Thus, the robustness, high efficiency, and high accuracy of our strategy for selection-free, simultaneous generation of single, double, and triple knock-in clone collections were demonstrated.

**Versatility of the LoAD system**. To further expand the application of the LoAD system, we additionally examined the enhancement of other types of genome editing by accumulating various effector proteins. First, we focused on erroneous end-joining-mediated short deletion without a targeting donor (Fig. 4a). Since a previous study revealed that the overexpression of Trex2 resulted in higher mutagenic efficiency[21], we observed the effects of Trex2 and MS2-Trex2, as well as Exo1, MS2-Exo1, CtIP, and MS2-CtIP, at the *LMNB1* locus. Restriction fragment length polymorphism (RFLP) analysis[22] revealed that the LoAD-ing of MS2-Trex2 resulted in higher efficiency of short deletions compared with mock or CtIP-overexpressing samples (Fig. 4b, Supplementary Fig. 18a). A similar effect was also observed at the *RPL11* locus (Fig. 4c, Supplementary Fig. 18b). Consistent with the results of the RFLP analyses, subcloned sequencing analysis revealed that the rates of short deletion among the total number of indel alleles were 25%, 66.7%, and 90.9% in mock, Trex2, and MS2-Trex2 samples, respectively (Fig. 4d, Supplementary Fig. 18c). Next, we investigated whether the frequency of chromosomal deletions induced by double cutting[23] was affected by the LoADing of certain molecules (Fig. 4e, Supplementary Fig. 19a). Consistent with the previous report[24], we found that LoADed MS2-Trex2 strongly inhibited chromosomal deletions in two different sgRNA pairs (Fig. 4f, g, Supplementary Fig. 19c, d); whereas, the independent mutation efficiency was enhanced (Fig. 4h, Supplementary Fig. 19b, e, f). Finally, we found the common knock-in-enhancing effect of LoADed MS2-CtIP on

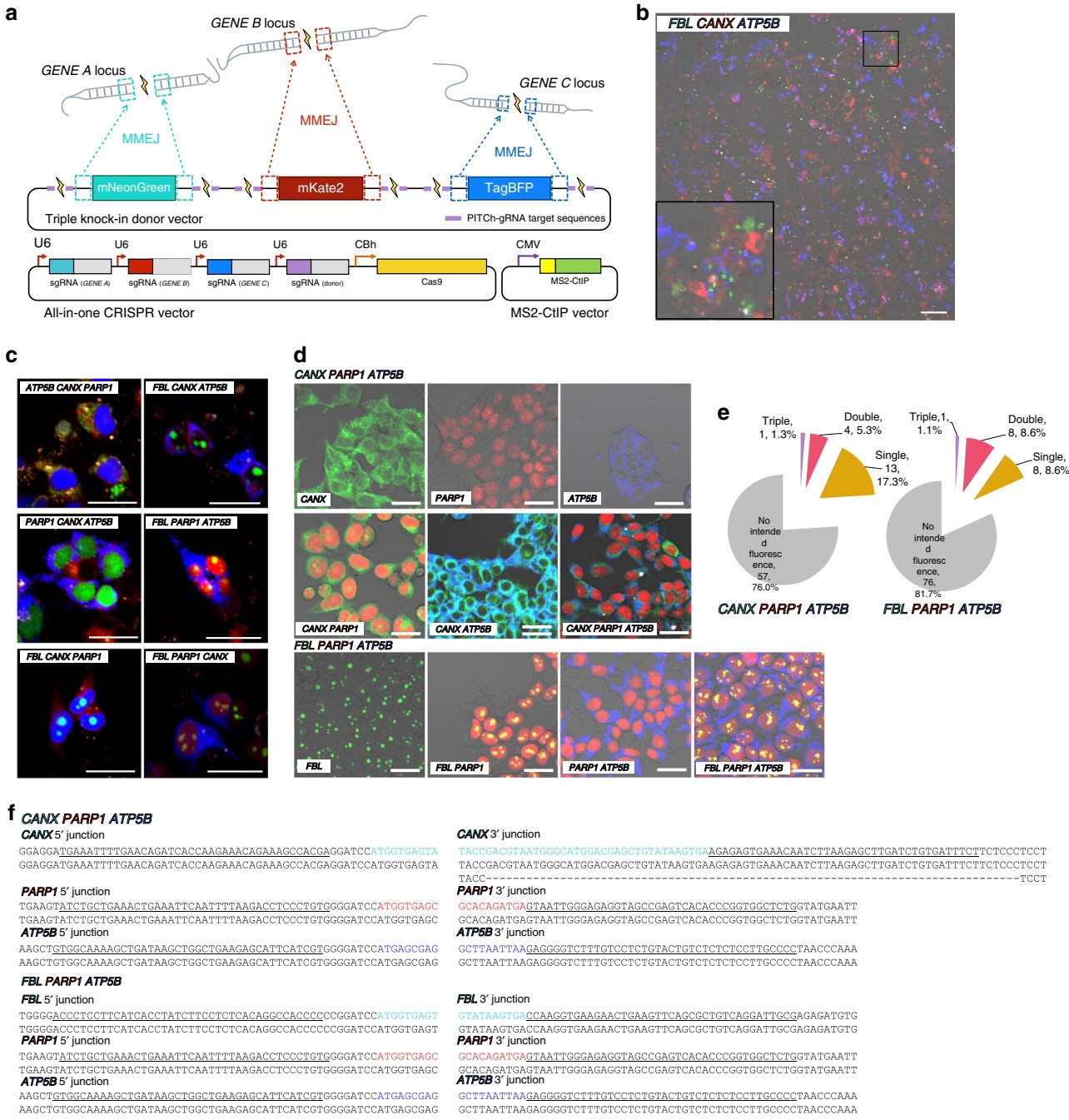

**Fig. 3** Simultaneous generation of multicolored cell collections. **a** Schematic of triple gene knock-in with the enhanced PITCh system. Three independent loci were targeted by a single-donor vector containing green, red, and blue fluorescent protein genes flanked by specific microhomologous arms for each gene locus. Triple gene knock-in was achieved by co-transfection of the integrated donor vector, all-in-one CRISPR vector, and MS2-CtIP vector. U6, human U6 promoter; CBh, chicken beta-actin short promoter; CMV, cytomegalovirus promoter. **b** A merged low-power image of heterogeneous cell populations transfected with triple knock-in vectors illustrated in Fig. 1a, targeting *FBL*, *CANX*, and *ATP5B* genes. Differential interference contrast and green, red, and blue fluorescence images were merged. A magnified view of the region boxed in black is displayed in the bottom-left corner. Bar, 100 μm. **c** Merged high-power images of triple knock-in cells transfected with knock-in vectors with various combinations of target genes with fluorescent protein genes. Green, red, and blue fluorescence images were merged. Bar, 30 μm. **d** Examples of the fluorescence images of various single, double, and triple knock-in cell clones simultaneously generated by a single transfection and single-cell cloning for each combination. Bar, 30 μm. **e** Numbers and percentages of single, double, and triple gene knock-in clones in the two types of triple gene knock-in. **f** Sanger sequencing of knock-in junctions of triple knock-in clones. The intended knock-in sequences are shown at the top of each sequence. Green, red, and blue letters indicate the coding sequences of mNeonGreen, mKate2, and TagBFP, respectively. Dashes indicate deletions. Underlines indicate microhomologies

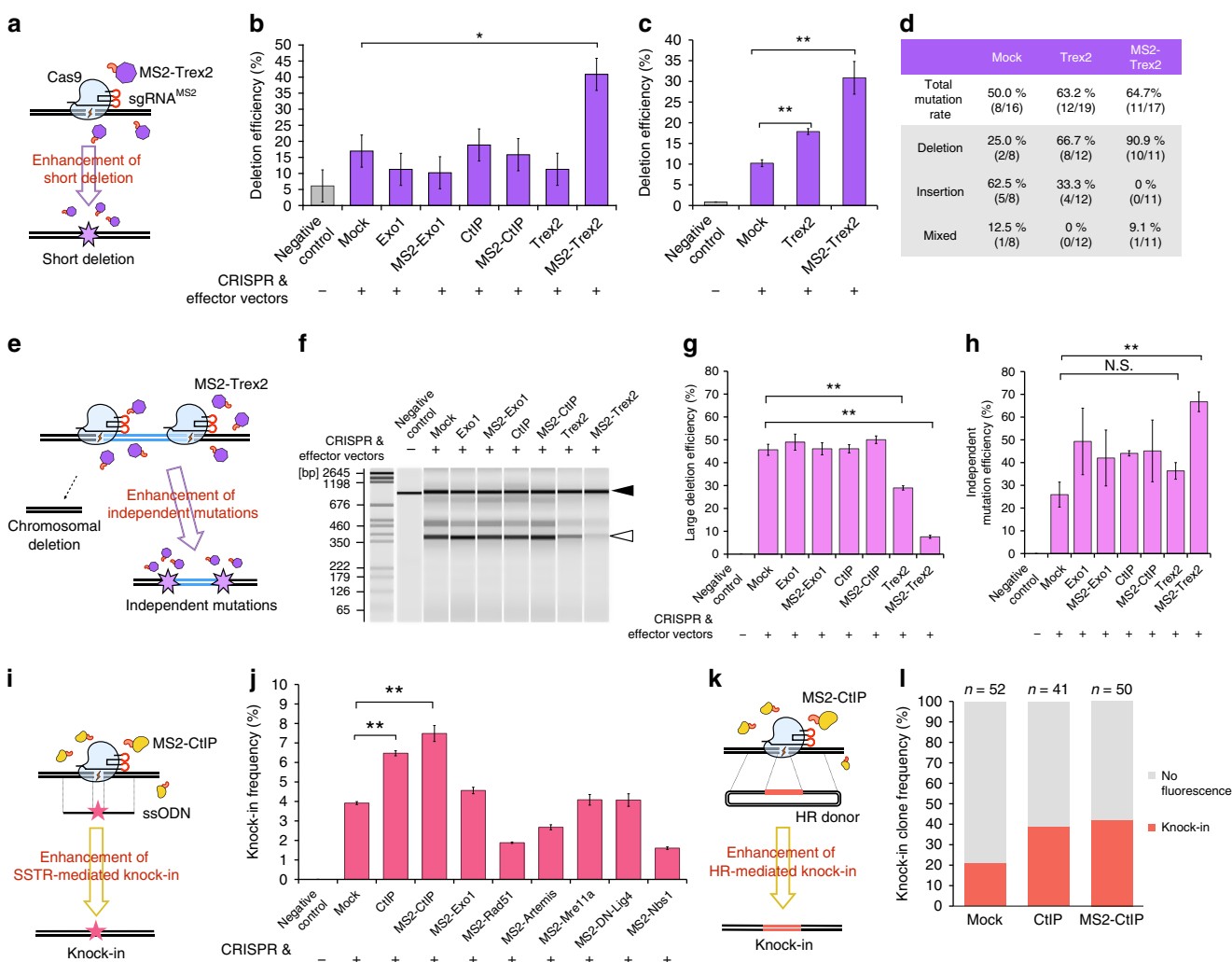

**Fig. 4** LoAD system can affect various genome editing outcomes. **a** Schematic of the enhancement of short deletion with MS2-Trex2 LoADing. **b** Frequency of short deletion at the *LMNB1* locus detected by RFLP analysis. Data are expressed as means ± s.e.m. ($n = 3$). *$P < 0.05$ (Student's $t$-test). **c** Frequency of short deletion at the *RPL11* locus detected by RFLP analysis. Data are expressed as means ± s.e.m. ($n = 3$). **$P < 0.01$ (Student's $t$-test). **d** Subcloned sequencing analysis revealed that the LoADing of MS2-Trex2 resulted in high-frequency deletions. **e** Schematic of the enhancement of independent mutations and the suppression of chromosomal deletion with MS2-Trex2 LoADing. **f** Pseudo-gel image of the out–out PCR products at *APC* locus 1, illustrated in Supplementary Figure 19a. Black and white arrowheads indicate the expected positions of the wild type and chromosomally deleted PCR products, respectively. **g** Percentages of chromosomally deleted alleles among all the out–out PCR products at *APC* locus 1. Data are expressed as means ± s.e.m. ($n = 3$). **$P < 0.01$ (Student's $t$-test). N.S., not significant. **h** Percentages of independently mutated alleles among the full-length PCR products indicated by the black arrowhead in Fig. 4f. The mutations were detected by genomic cleavage detection assay[36]. Pseudo-gel image of this assay is shown in Supplementary Figure 19b. Data are expressed as means ± s.e.m. ($n = 3$). **$P < 0.01$ (Student's $t$-test). **i** Schematic of the enhancement of ssODN knock-in mediated by single-strand template repair (SSTR) with MS2-CtIP LoADing. **j** Efficiencies of ssODN knock-in detected by RFLP analysis, related to Supplementary Figure 20. Data are expressed as means ± s.e.m. ($n = 3$). **$P < 0.01$ (Student's $t$-test). **k** Schematic of the enhancement of plasmid knock-in mediated by homologous recombination (HR) with MS2-CtIP LoADing. **l** Efficiencies of HR-mediated knock-in, related to Supplementary Figure 21. The knock-in rate was calculated as the percentages of knock-in clones among the total number of clones

single-strand template repair[25] (Fig. 4i, j, Supplementary Fig. 20) and HR repair pathways (Fig. 4k, l, Supplementary Fig. 21).

## Discussion
In this study, we established a system of biased genome editing by recruiting specific repair molecules such as CtIP at the DSB site, and then demonstrated the simultaneous generation of multi-colored cell collections, as well as showing other applications including the efficient independent mutagenesis with the inhibition of chromosomal deletions.

In particular, consistent with the recent report describing the knock-in-enhancing effect by Cas9 fused with CtIP (Cas9-CtIP)[14], the enhancement of MMEJ-, SSTR-, and HR-mediated knock-in

was achieved by LoADing MS2-CtIP. The enhancement of MMEJ-mediated knock-in by MS2-CtIP LoADing should become popular because the MMEJ-dependent strategy has been utilized in various applications including scarless disease modeling in human-induced pluripotent stem cells[26], in vivo gene knock-in[27], gene knock-in in neurons[28], epitope tagging for chromatin immunoprecipitation (ChIP)-seq analysis[29], and generation of knock-in mice containing exogenous gene cassette[9], floxed allele[9], or multiple SNPs spanning multiple exons[30]. In addition, a recent screening-based approach for the investigation of genetic interactions[31] revealed the need for a high-throughput strategy for generating cell clone libraries harboring various combinations of gene knock-in alleles. Our system will be an adequate technology

for such high-throughput cell engineering. On another front, however, potential risk of increasing off-target mutations by MS2-CtIP LoADing should be carefully examined in the future study, although, we showed the risk of off-target integration was minimum even when MS2-CtIP was LoADed.

In the analysis of CtIP mutants, we could not obtain any superior results compared with wild-type MS2-CtIP in HEK293T cells. The 3KR mutant, circumventing the requirement of SIRT6[17], did not alter the knock-in-enhancing effect. The S327E/T847E mutations, which mimicked phosphorylation[15,16], decreased the activity. However, these mutants might increase the activity in other cells with low SIRT6 expression and/or poor phosphorylation of CtIP. On the other hand, all the truncated HE mutants[14] with or without additional amino acid changes showed comparable knock-in efficiency to mock-transfected control, although Charpentier et al. showed that Cas9 fused with HE mutant (Cas9-HE) resulted in higher HDR efficiency compared to that of Cas9[14]. In addition, full-length MS2-CtIP harboring either phosphorylation-blocking (3A) or phosphorylation-mimicked (3E) mutation positioned within the HE region did not show any change in knock-in efficiency. These findings suggest that HE region is not functional in our LoADed PITCh system.

Collectively, our results demonstrate the high-throughput generation of gene knock-in cell collections, which can contribute not only to the production of multiplexed reporter cells, but also to the simultaneous generation and evaluation of multiple polymorphisms spanning several gene loci for analyzing phenotypic variations or disease severity caused by multifactorial inheritance. Although, the utility of our LoAD system was demonstrated, the examples of its application presented in this paper were based on the screening of only a small number of DSB repair molecules. Further possibilities of the LoAD system will be revealed by large-scale effector screening, contributing to much more biased, controllable, and precise genome engineering.

## Methods

**Plasmid construction and oligonucleotide preparation**. The all-in-one CRISPR-Cas9 vectors expressing Cas9 nuclease and single or multiple sgRNAs were constructed using the Multiplex CRISPR/Cas9 Assembly System Kit (Kit #1000000055, Addgene)[20] with some modifications. Briefly, oligonucleotides for sgRNA templates were synthesized, annealed, and inserted into the modified pX330A and pX330S vectors. Golden Gate assembly was used to assemble the constructed vectors into the all-in-one CRISPR/Cas9 vectors. For the addition of MS2 stem loops to sgRNA scaffold, the sgRNA expression cassette of the pX330A and pX330S vectors was replaced with that of the sgRNA(MS2) cloning backbone vector (Plasmid #61424, Addgene). The oligonucleotides used for the sgRNA template are listed in Supplementary Data 1.

The effector vectors without MS2 coat protein were constructed using an RT-PCR and In-Fusion cloning method (Takara)[9]. For the CtIP cloning, the CtIP-vector-F/R and CtIP-insert-F/R primers listed in Supplementary Data 1 were used. For the effector vectors with MS2 coat protein, we first constructed MS2-Exo1 vector by the addition of MS2-NLS obtained from the MS2-P65-HSF1_GFP vector (Plasmid #61423, Addgene) to the Exo1 overexpression vector by the In-Fusion cloning method using the primers listed in Supplementary Data 1. Then, the coding sequence of Exo1 was replaced with other effector cDNAs using the primers listed in Supplementary Data 1.

The MS2-CtIP vectors containing either 3KR or S327E/T847E mutations, named MS2-CtIP(3KR) vector and MS2-CtIP(S327E/T847E) vector, respectively, were constructed by In-Fusion cloning method templated with the MS2-CtIP vector using the following primers: CtIP-3KR-vector, CtIP-3KR-insert-1, and CtIP-3KR-insert-2 for MS2-CtIP(3KR); and CtIP-S327E/T847E-vector and CtIP-S327E/T847E-insert for MS2-CtIP(S327E/T847E). For the construction of the MS2-CtIP vector containing both 3KR and S327E/T847E mutations, named MS2-CtIP(S327E/T847E, 3KR) vector, the insert DNA fragment was amplified from the MS2-CtIP(3KR) vector using CtIP-3KR-S327E/T847E-insert primers, the vector backbone was amplified from the MS2-CtIP(S327E/T847E) vector using CtIP-3KR-S327E/T847E-vector primers, and then they were assembled by In-Fusion cloning method. The C-terminally truncated MS2-CtIP mutant vector, named MS2-CtIPHE vector, was constructed by the inverse PCR and In-Fusion cloning method using CtIP-HE primers. For the construction of the MS2-CtIP and MS2-CtIPHE vectors containing either 3A or 3E mutations, named MS2-CtIP(3A),

MS2-CtIP(3E), MS2-CtIPHE(3A), and MS2-CtIPHE(3E), respectively, the DNA fragments encoding HE(3A) and HE(3E) were synthesized with gBlocks (IDT), and inserted into the MS2-CtIP or MS2-CtIPHE vector by In-Fusion cloning method with the PCR-amplified vector backbones generated as follows: for the MS2-CtIP (3A) and MS2-CtIP(3E) vectors, the MS2-CtIP vector was used as a template and CtIP-gBlocks primers were used; and for the MS2-CtIPHE(3A) and MS2-CtIP(3E) vectors, the MS2-CtIPHE vector was used as a template and CtIP-HE-gBlocks primers were used. The sequence of the primers and gBlocks fragments is shown in Supplementary Data 1.

The PITCh donor vectors carrying mNeonGreen were constructed by PCR and TA-cloning with TArget Clone™ -Plus- (Toyobo) using the primers listed in Supplementary Data 1, except the donor for the FBL locus. For the construction of the FBL-mNeonGreen donor vector, the T2A-Puro[R] sequence was removed from the PITCh(gRNA-s1)-FBL[9]. The PITCh donor vectors carrying 3× Flag tag, donor vectors for triple knock-in, and the HR donor vector were constructed by PCR and In-Fusion cloning (Takara) using the primers listed in Supplementary Data 1. The PITCh donor vectors for triple knock-in were constructed by PCR and In-Fusion cloning using the primers listed in Supplementary Data 1, as illustrated in Supplementary Fig. 14a.

The oligonucleotide for SSTR-mediated knock-in was purchased from IDT. The sequence is shown in Supplementary Data 1.

The plasmid sequences of CtIP vector, MS2-CtIP vector, and the PITCh donor vectors for Fig. 2d–f are shown in Supplementary Data 4.

**Cell culture and transfection**. HEK293T and HeLa cells, obtained from ATCC, were maintained in Dulbecco's modified Eagle's medium supplemented with 10% fetal bovine serum (FBS). CHO-K1 cells, obtained from Riken BRC Cell Bank, were maintained in Ham's F12 medium supplemented with 10% FBS. All cell lines were tested negatively for mycoplasma contamination using e-Myco Mycoplasma PCR Detection Kit (iNtRON Biotechnology). Human cells (HEK293T and HeLa cells) were authenticated by short tandem repeat analysis (Takara).

Lipofectamine LTX (Life Technologies) and serum-free DMEM were used to transfect the plasmids in accordance with the supplier's protocol, except the SSTR-mediated knock-in. For the SSTR-mediated knock-in, the 4D-Nucleofector system was used with the solution SF under the program CM-130, in accordance with the supplier's protocol. The amounts of plasmids and oligonucleotides, cell numbers, and plates used were as follows: for the FACS analysis and fluorescence observation of mNeonGreen knock-in HEK293T cells at the CANX locus, 200 ng of plasmids in total (PITCh donor vector, CRISPR vector, and mock-overexpression/LoADing vector) into $4.5 \times 10^4$ cells using a 96-well plate; for the FACS analysis and fluorescence observation of mNeonGreen knock-in HEK293T cells at the ATP5B, PARP1, FBL, and RPL11 loci, 200 ng of plasmids in total (PITCh donor vector, CRISPR vector, and mock/CtIP/MS2-CtIP vector) into $7.5 \times 10^4$ cells using a 96-well plate; for the mNeonGreen knock-in at the ATP5B locus, followed by single-cell cloning, genomic out–out PCR, and Southern blotting, 200 ng of plasmids in total (PITCh donor vector, CRISPR vector, and mock/MS2-CtIP vector) into $5.0 \times 10^4$ cells using a 96-well plate; for the cytotoxicity analysis, 1, 3.5, 12.2, 42.9, and 150 ng each for mock/CtIP/MS2-CtIP vectors, or 150 ng for the positive control ZFN vector, pSTL-ZFA36[32], and 150 ng for mCherry vector into $5 \times 10^4$ cells using a 96-well plate; for the γ-H2AX immunostaining, 200 ng of plasmids in total (mCherry vector and mock/CtIP/MS2-CtIP vector) into $5.0 \times 10^4$ cells using a 96-well plate; for the characterization of MS2-CtIP mutants with mNeonGreen knock-in at the CANX locus, 200 ng of plasmids in total (PITCh donor vector, CRISPR vector, and mock/CtIP/MS2-CtIP/MS2-CtIP mutant vector) into $5.0 \times 10^4$ cells using a 96-well plate, with transferring the cells to a 24-well plate at 24 h post transfection; for the genomic PCR and NGS analysis of 3× Flag tag knock-in cells at the ATP5B, PARP1, and RPL11 loci, 200 ng of plasmids in total (PITCh donor vector, CRISPR vector, and mock/CtIP/MS2-CtIP vector) into $4.5 \times 10^4$ cells using a 96-well plate; for the single-cell cloning and fluorescence observation of triple knock-in cells, 400 ng of plasmids in total (PITCh donor vector, CRISPR vector, and mock/CtIP/MS2-CtIP vector) into $4.5 \times 10^4$ cells using a 96-well plate; for the genomic PCR, followed by RFLP, Cel-I, and subcloned sequencing analyses at the LMNB1, RPL11, and APC loci, 100 ng of plasmids in total (CRISPR vector and mock/overexpression/LoADing vector) into $1.5 \times 10^4$ cells using a 96-well plate; for the genomic PCR, followed by RFLP analysis of SSTR-mediated knock-in cells at the HPRT1 locus, 2000 ng of plasmids (CRISPR vector and mock/overexpression/LoADing vector) and 100 pmol ssODN into $2.0 \times 10^5$ cells using a 24-well plate; and for the single-cell cloning and fluorescence observation of HR-mediated knock-in cells, 200 ng of plasmids in total (PITCh donor vector, CRISPR vector, and mock/CtIP/MS2-CtIP vector) into $7.5 \times 10^4$ cells using a 96-well plate; for the FACS analysis and fluorescence observation of mNeonGreen knock-in CHO-K1 cells at the Canx and Atp5b loci, 200 ng of plasmids in total (PITCh donor vector, CRISPR vector, and mock/CtIP/MS2-CtIP vector) into $6.0 \times 10^4$ cells using a 96-well plate; for the FACS analysis and fluorescence observation of mNeonGreen knock-in HeLa cells at the PARP1 locus, 200 ng of plasmids in total (PITCh donor vector, CRISPR vector, and mock/CtIP/MS2-CtIP vector) into $4.0 \times 10^4$ cells using a 96-well plate.

**FACS analysis**. The knock-in cells and the cells for the cytotoxicity analysis were collected at 72 h and 5 d post transfection, respectively, suspended in PBS, and

filtered with a Flowmi Tip Strainer (Bel-Art Products). The number of cells with green and red fluorescence was counted using a BD FACSCalibur 4A (BD Biosciences) with a 488-nm laser and the corresponding fluorescence filters (FL1 for green, FL2 for red). After the preliminary FSC/SSC gating, up to 10,000 cells were analyzed for each sample. The data were analyzed using BD CellQuest Pro ver. 5.2 (BD Biosciences). For the cytotoxicity analysis, cell survival rates were determined as the percentages of the number of mCherry-positive cells in total cell counts[9,33].

**Single-cell cloning**. For the triple knock-in and HR knock-in, cells were collected at 72 h post transfection, suspended in PBS, and filtered with a Flowmi Tip Strainer. Single cells were sorted using a PERFLOW® Sort (Furukawa Electric) without fluorescence filters, collected in 96-well plates containing DMEM medium supplemented with 10% FBS, and cultured for 1–3 weeks until the colonies formed. For the mNeonGreen knock-in, cells were collected at 72 h post transfection, adjusted to 1.5 cells per 100 µl by adding DMEM, and moved 100 µl of adjusted cells to each well of a 96-well plate.

**Fluorescence imaging**. Cells were moved to collagen-coated glass-bottomed 24-well plates at 72 h post transfection, cultured for an additional 48 h, and fixed with 4% paraformaldehyde in PBS. Fluorescence was observed and cell images were captured with 405-, 473-, and 594-nm lasers using a confocal laser-scanning microscope (Olympus FV-1000D) with the software FV10-ASW ver. 04.02.02.09 (Olympus).

**γ-H2AX immunostaining**. Cells were moved to collagen-coated glass-bottomed 24-well plates at 24 h post transfection, and cultured for an additional 48 h. DSB induction and γ-H2AX immunostaining were performed using an OxiSelect™ DNA DSB Staining Kit (CELL BIOLABS, INC.), according to the manufacturer's protocol. Briefly, the cells were treated with 1 µM etoposide (Part No. 232103) in the medium for 1 h at 37 °C, fixed with 4% paraformaldehyde in PBS, and permeabilized with 90% methanol. Subsequently, the cells were blocked with 1% BSA in PBS for 1 h at room temperature, and then incubated with Anti-Phospho-Histone H2A.X(Ser 139) Antibody (Part No. 232101) at a 1:100 dilution ratio for overnight at 4 °C. After washing with PBS with Tween 20 (PBS-T), the cells were stained with Secondary Antibody, FITC Conjugate (Part No. 232102) at a 1:100 dilution ratio for 1 h at room temperature. After washing with PBS-T, nuclei were counterstained with DAPI. The fluorescence images of DAPI, mCherry, and γ-H2AX were obtained using a confocal laser-scanning microscope (Olympus FV-1000D). For the imaging analysis, Fiji software (ver. 2.0.0-rc-67/1.52d (https://fiji.sc/) was used to determine the areas of red (mCherry) and green (FITC) fluorescence from the captured images. The γ-H2AX activation of the transfected cells was quantified as the percentages of the FITC-positive areas in the mCherry-positive areas.

**Genomic PCR and direct sequencing of knock-in junctions**. Genomic DNA was extracted using a DNeasy Blood and Tissue kit (Qiagen) from the knock-in cells collected at 72 h post transfection. Genomic PCR was performed using PrimeSTAR Max DNA Polymerase (Takara) or KOD FX Neo (Toyobo) with the primers listed in Supplementary Data 1. For the allele number analysis, the PCR products were run on a 2% (wt/vol) agarose gel. Direct sequencing analysis of knock-in junctions was performed by FASMAC using an ABI 3130xl Genetic Analyzer (Life Technologies) with a BigDye Terminator v3.1 Cycle Sequencing Kit (Life Technologies). The sequence data were collected using the software 3130 Series Data Collection Software 4 (Life Technologies).

**Southern blot analysis**. A total of 5-µg aliquots of genomic DNA were digested with EcoRV, PvuII, and SacI, and resolved on a 0.7% agarose gel. $^{32}$P-labeled DNA probes were made by PCR using Ex Taq polymerase (Takara) and [alpha-$^{32}$P] dATP (Perkin Elmer) with primers listed in Supplementary Data 1. Membrane transfer with Hybond-N+ (GE Healthcare), ultraviolet cross-linking (120 mJ cm$^{-2}$), pre-hybridization and hybridization were performed according to the instructions for ExpressHyb Hybridization Solution (Takara). The radio-isotope signal was detected using a bioimaging analyzer BAS-2500 (Fujifilm).

**NGS and data analysis**. Genomic DNA was extracted using a DNeasy Blood and Tissue kit from the knock-in cells collected at 72 h post transfection. Preparation of libraries and NGS were performed by FASMAC. Briefly, NGS libraries were prepared via two-step PCR using the primers listed in Supplementary Data 1. The PCR amplification was performed using PrimeSTAR GXL DNA Polymerase (TakaRa) and the PCR products were purified using Agencourt AMPure XP (Beckman Coulter). Note that the PCR products amplified using PrimeSTAR Max DNA Polymerase were used as templates in the first-step PCR of *PARP1* samples instead of genomic DNA because genomic PCR amplification failed in the *PARP1* samples. NGS was performed using a MiSeq (Illumina). The PCR products were also analyzed by microchip electrophoresis using MultiNA ™ (hereinafter referred to as "MultiNA") (Shimadzu) with the software MultiNA ver. 1.11 (Shimadzu).

The NGS data were analyzed using CRISPResso ver. 1.0.0[34] (Fig. 2b) or Cas-Analyzer[35] (Fig. 2c, d, Supplementary Fig. 11–13). The parameters used for

CRISPResso analysis are described below. The parameters used for Cas-Analyzer were as follows: comparison range, 70; minimum frequency, 0; and WT marker, 5. The exported results of CRISPResso and Cas-Analyzer are described in Supplementary Data 2 and 3, respectively.

**Parameters used for the CRISPResso analysis**. The following parameters were used for the CRISPResso analysis.

```
RPL11:
-a
ggggtagatggaaggggaggaatatggcttttaacaggagcc
ccctttctcagatgatagtgcagttcagcacagtgtaaaaac
cagccagcttcctatttagtccagaaaaggatgggattcaga
gcccaagttcatgtatcaatcagatgtgaattctcaaaagtt
agccattgctgcaatctctgctgttgcctcctgttctgaaaa
aattaaatctcttctcttcagtatgatgggatcatccttcc
tggcaaataaattcccgtttctatccaaaagagcaataaaaa
gttttcagtgaaatgtgcaattctgttgtgtgttctgtgaaa
ggatcctggc
-g
cggggaatttatttgccagga
-e
ggggtagatggaaggggaggaatatggcttttaacaggagcc
ccctttctcagatgatagtgcagttcagcacagtgtaaaaac
cagccagcttcctatttagtccagaaaaggatgggattcaga
gcccaagttcatgtatcaatcagatgtgaattctcaaaagtt
agccattgctgcaatctctgctgttgcctcctgttctgaaaa
aattaaatctcttctctttcagtatgatgggatcatccttcc
gGGATCCGACTATAAGGACCACGACGGAGACTACAAGGATCA
TGATATTGATTACAAAGACGATGACGATAAGTAAtggcaaat
aaattcccgtttctatccaaaagagcaataaaaagttttcagt
gaaatgtgcaattctgttgtgtgttctgtgaaaggatcctggc
-d
gGGATCCGACTATAAGGACCACGACGGAGACTACAAGGATCAT
GATATTGATTACAAAGACGATGACGATAAGTAA
--hide_mutations_outside_window_NHEJ
--hdr_perfect_alignment_threshold 99.5
PARP1:
-a
gcacatgtacataccctctgttgtatggctgttggctccttaa
caagcttcccctcaggtacattgtctatgatattgctcaggta
aatctgaagtatctgctgaaactgaaattcaattttaagacct
ccctgtgtaattgggagaggtagccgagtcacacccggtggc
tctggtatgaattcacccgaagcgcttctgcaccaactcacct
ggccgctaagttgctgatgggtagtacctgtactaaaccacct
cagaaaggattttacagaaacgtgttaaaggttttctctaact
tctcaagtcccttgttttgtgttgtgtctgtggggaggggttg
ttttg
-g
ctacctctcccaattaccac
-e
Gcacatgtacataccctctgttgtatggctgttggctccttaa
caagcttcccctcaggtacattgtctatgatattgctcaggta
aatctgaagtatctgctgaaactgaaattcaattttaagacct
ccctgtggGGATCCGACTATAAGGACCACGACGGAGACTACAAG
GATCATGATATTGATTACAAAGACGATGACGATAAGTAAgtaat
tgggagaggtagccgagtcacacccggtggctctggtatgaatt
cacccgaagcgcttctgcaccaactcacctggccgctaagttgc
tgatgggtagtacctgtactaaaccacctcagaaaggattttta
cagaaacgtgttaaaggttttctctaacttctcaagtcccttg
ttttgtgttgtgtctgtggggaggggttgttttg
-d
gGGATCCGACTATAAGGACCACGACGGAGACTACAAGGATCAT
GATATTGATTACAAAGACGATGACGATAAGTAA
--hide_mutations_outside_window_NHEJ
--hdr_perfect_alignment_threshold 99.5
ATP5B:
-a
ttacatgatgcagaaagttgatatccctccgcttcttactctt
tttttttttctcccccatcatacaggtgaatatgaccatctcc
cagaacaggccttctatatggtgtgggacccattgaagaagctgt
ggcaaaagctgataagctggctgaagagcattcatcgtgaggg
gtctttgtcctctgtactgtctctctccttgcccctaacccaa
aaagcttcattttctgtgtaggctgcacaagagcccttgatt
gaagatatattcttctgaacagtatttaaggtttccaataa
aatgtacaccctcagaatttgtctgattctcttggtt
-g
tgaagagcattcatcgtgag
-e ttacatgatgcagaaagttgatatccctccgcttcttact
cttttttttttctcccccatcatacaggtgaatatgac
catctcccagaacaggccttctatatggtgtgggacccattgaa
```

```
gaagctgtggcaaaagctgataagctggctgaagagcatt
catcgtggGGATCCGACTATAAGGACCACGACGGAGACTACAAG
GATCATGATATTGATTACAAAGACGATGACGATAAGTAA
gaggggtctttgtcctctgtactgtctctctccttgcccc
taacccaaaaagcttcattttttctgtgtaggctgcacaa
gagccttgattgaagatatattctttctgaacagtatt
taaggtttccaataaaatgtacacccctcagaatttgtct
gattctcttggtt
-d
ggGGATCCGACTATAAGGACCACGACGGAGACTACAAGGATCAT
GATATTGATTACAAAGACGATGACGATAAGTAA
--hide_mutations_outside_window_NHEJ
--hdr_perfect_alignment_threshold 99.5
```

**Genotyping of the cells other than PITCh and HR samples**. For the genotyping of the cells other than PITCh and HR knock-in samples, the cells were collected at 72 h post transfection. The templates for PCR amplification were prepared using the Cell Lysis Buffer in a GeneArt Genomic Cleavage Detection Kit (Life Technologies), in accordance with the manufacturer's instructions. The PCR amplification was performed using the primers listed in Supplementary Data 1. The pseudo-gel images were obtained and quantification of each amplicon was performed using MultiNA with the software MultiNA ver. 1.11. The RFLP analyses were performed using AluI, HpyAV, and AflII at the LMNB1, RPL11, and HPRT1 loci, respectively. The genomic cleavage detection assay was performed using a GeneArt Genomic Cleavage Detection Kit, in accordance with the manufacturer's instructions. Bacterial cloning of the PCR products was performed using TArget Clone™ -Plus-. Sanger sequencing was performed by FASMAC or using an ABI 3130xl Genetic Analyzer with a BigDye Terminator v3.1 Cycle Sequencing Kit. The sequence data were collected using the software 3130 Series Data Collection Software 4.

**Statistical analysis**. Statistical analysis was performed using the R software ver. 3.4.1. All data are presented as the mean ± s.e.m. All measurements were taken from distinct samples. Statistical significance was determined by Student's $t$-test.

**Data availability**. The authors declare that all data supporting the findings of this study are available within the paper and its Supplementary Information files. The plasmid sequences shown in Supplementary Data 4 were also deposited in NCBI via DDBJ. The accession codes are LC307119–LC307122. The NGS data were submitted to NCBI via DDBJ and are accessible with BioProject ID: PRJDB7211.

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

## Acknowledgements

We would like to acknowledge the technical assistance of Mitsumasa Takenaga (Hiroshima University) for plasmid construction, Yusaku Wada (FASMAC Co. Ltd.) for next-generation sequencing, Toru Ezure (Shimadzu Co. Ltd.) and Ken Tsukii (Furukawa Electric Co. Ltd.) for cell sorting using a PERFLOW Sort, and Tatsuo Mukai for the guidance of bioinformatics analysis. We also thank Feng Zhang (Broad Institute) for the provision of reagents through Addgene. The mNeonGreen cDNA was used under a license agreement with Allele Biotechnology and Pharmaceuticals, Inc. This work was funded in part by grants from the Japan Society for the Promotion of Science (17K15045 to S.N., 16K18478 to T.S., and 17H01409 to T.Y. and T.S.) and the Japan Agency for Medical Research and Development (AMED) (Research Program on Hepatitis to T.Y.).

## Author contributions

T.S. and S.N. conceived and designed the study. S.N. performed the majority of experiments. K.M. performed experiments other than PITCh applications. A.K. performed γ-H2AX staining. K.N. analyzed the data. N.S. performed Southern blotting. T.A.

and K.T. provided instructions. T.Y. supervised the study. T.S. wrote the manuscript with feedback from all authors.

## Additional information

**Competing interests:** The authors declare no competing interests.

