## [Peer Review File · Nature Communications]

Reviewers' comments:

Reviewer #1 (Remarks to the Author):

In this paper the authors explore the effect of targeting the resection factor CtIP through a fusion to the RNA-binding protein MS2 on CRISPR-Cas9 MMEJ mediated knock ins (KI). By several approaches, the authors observe a consistent increase in the efficiency in single, but also multiple KIs. Also, they observed a reduction in INDEL and imprecise KI by targeting CtIP to the relevant loci.

In general, the data seem solid and the increase the authors observe in KI efficiency, around two-fold, make this study of interest. However, before supporting the publication I find to major concerns that the authors should address:

1. As it is indeed stated in the discussion, one major caveat is that there is no analysis of the off-targets integrations caused by MS2-CtIP LoADing. Despite the claim that this should be pursued in a future study, I find it unavoidable to tackle the issue here.
2. Also, there is no study of the possible toxic effect of the MS2-CtIP expression. Is cell viability compromised in any form? Are cells expressing such construct able to repair DNA breaks efficiently? Is DNA resection upon exogenous damage normal on those cells? Only if this two points are correctly addressed I will support publication of the manuscript.

Reviewer #2 (Remarks to the Author):

The recent advance in CRISPR-mediated novel precision genome editing technologies are revolutionizing our approach to studying gene function. Despite the huge success of modification of genome by HDR- and NHEJ-mediated DSB repair, the efficiency of knock-in is still relatively low. Sakuma and Yamamoto et al first demonstrated the MMEJ-mediated integration of donor DNA in vitro and in vivo (termed PITCh, Precise Integration into Target Chromosome) in 2014. A follow up study in 2016 further improved the efficiency with new CRIS-PITCh (v2) system. In this manuscript, Nakade et al described a novel approach to enhance MMEJ-mediated knock-in using LoAD system which increases the local concentration of CtIP/RBBP8. In a proof-of-concept study, cell lines with multiplexed reporter (fluorescent protein) were generated by this new system. This is a nicely crafted paper with clean and convincing data. It will be of interest to a wide audience. I only have a few questions/concerns:

1. The authors claim that "...previous studies never achieved simultaneous generation of various patterns or combinations of multiple reporters in multiple genomic loci" (page 2, line 6). Actually, Zhong et al generated two reporters in two genes (Tet1-EGFP and Tet3-ECFP) via HDR-mediated knock-in in mouse androgenetic haploid embryonic stem cells (PMID: 26165924, Figure S5). It would be nice to cite this paper and mention their work.
2. Figure 3e. for knock-in genes, the authors did not mention whether they are homozygous or heterozygous.
3. Although it is not the central focus of the paper, for Figure 4i, the enhancement of ssODN knock-in with LoADing, did the author also test multiple sites ssODN knock-in? If it works, people would use it for loxP sites knock-in with two ssODN (PMID: 23992847).
4. Phosphorylation of CtIP is required for the activation of DSB processing (PMID: 23273981). It would be great if the author can test mutant CtIP forms (e.g. constitutively active, phosphorylation site-mutated, etc.) in the LoAD system.

Response to Reviewer #1

We appreciate the great help from the reviewer for the improvement of our paper. A summary of the revisions made in accordance with the comments from the reviewer, together with our point-by-point response to these comments, is shown below. The valuable suggestions of the reviewer are also listed again for explanation.

Remarks to the Author

In this paper the authors explore the effect of targeting the resection factor CtIP through a fusion to the RNA-binding protein MS2 on CRISPR-Cas9 MMEJ mediated knock ins (KI). By several approaches, the authors observe a consistent increase in the efficiency in single, but also multiple KIs. Also, they observed a reduction in INDEL and imprecise KI by targeting CtIP to the relevant loci.

In general, the data seem solid and the increase the authors observe in KI efficiency, around two-fold, make this study of interest. However, before supporting the publication I find to major concerns that the authors should address:

1. As it is indeed stated in the discussion, one major caveat is that there is no analysis of the off-targets integrations caused by MS2-CtIP LoADing. Despite the claim that this should be pursued in a future study, I find it unavoidable to tackle the issue here.

Response: We appreciate this important indication. In accordance with the reviewer's indication, we additionally performed Southern blotting analysis to detect off-target integrants. The results were shown in newly added Supplementary Figure 8 and described in page 4 of the main text.

2. Also, there is no study of the possible toxic effect of the MS2-CtIP expression. Is cell viability compromised in any form? Are cells expressing such construct able to repair DNA breaks efficiently? Is DNA resection upon exogenous damage normal on those cells?

Response: We appreciate these important questions. Cell viability was tested by FACS analysis, and the capacity of DSB repair was analysed by γ -H2AX staining. The results were shown in newly added Supplementary Figure 9 and described in page 4 of the main text. Regarding DNA resection, we attempted to analyze it by RPA staining; however, it was difficult to determine the frequency of DNA resection because of high basal expression of RPA.

Only if this two points are correctly addressed I will support publication of the manuscript.

Response to Reviewer #2

We appreciate the great help from the reviewer for the improvement of our paper. A summary of the revisions made in accordance with the comments from the reviewer, together with our point-by-point response to these comments, is shown below. The valuable suggestions of the reviewers are also listed again for explanation.

Remarks to the Author

The recent advance in CRISPR-mediated novel precision genome editing technologies are revolutionizing our approach to studying gene function. Despite the huge success of modification of genome by HDR- and NHEJ-mediated DSB repair, the efficiency of knock-in is still relatively low. Sakuma and Yamamoto et al first demonstrated the MMEJ-mediated integration of donor DNA in vitro and in vivo (termed PITCh, Precise Integration into Target Chromosome) in 2014. A follow up study in 2016 further improved the efficiency with new CRIS-PITCh (v2) system. In this manuscript, Nakade et al described a novel approach to enhance MMEJ-mediated knock-in using LoAD system which increases the local concentration of CtIP/RBBP8. In a proof-of-concept study, cell lines with multiplexed reporter (fluorescent protein) were generated by this new system.

This is a nicely crafted paper with clean and convincing data. It will be of interest to a wide audience. I only have a few questions/concerns:

1. The authors claim that “...previous studies never achieved simultaneous generation of various patterns or combinations of multiple reporters in multiple genomic loci” (page 2, line 6). Actually, Zhong et al generated two reporters in two genes (Tet1-EGFP and Tet3-ECFP) via HDR-mediated knock-in in mouse androgenetic haploid embryonic stem cells (PMID: 26165924, Figure S5). It would be nice to cite this paper and mention their work.

Response: We appreciate this suggestion, and apologize for the oversight of previous achievement. In accordance with the reviewer’s suggestion, we have cited the Zhong’s paper and mentioned their work in page 2 of the main text.

2. Figure 3e. for knock-in genes, the authors did not mention whether they are homozygous or heterozygous.

Response: We appreciate this comment. In accordance with the reviewer’s comment, we performed hundreds of out-out PCR analyses against the genomic DNA samples collected from the clones shown in Figure 3e to determine homozygous or heterozygous knock-in.

Unfortunately, however, it was quite difficult to confirm homozygous or heterozygous for all the knock-in loci in all the clones due to the uncertainty of long PCR analysis and aneuploidy of HEK293T cells. Instead, we additionally established single knock-in clones and confirmed whether they were heterozygous or homozygous. The results were shown in newly added Supplementary Figure 7 and described in pages 3–4 of the main text.

3. Although it is not the central focus of the paper, for Figure 4i, the enhancement of ssODN knock-in with LoADing, did the author also test multiple sites ssODN knock-in? If it works, people would use it for loxP sites knock-in with two ssODN (PMID: 23992847).

Response: We appreciate this comment. As the reviewer pointed out, two ssODN-mediated flox generation in mouse zygotes was reported in the previous paper (PMID: 23992847). However, to the best of our knowledge, the applicability of this strategy has never been proven in cultured cells, possibly due to the low frequency of ssODN incorporation in cultured cells.

4. Phosphorylation of CtIP is required for the activation of DSB processing (PMID: 23273981). It would be great if the author can test mutant CtIP forms (e.g. constitutively active, phosphorylation site-mutated, etc.) in the LoAD system.

Response: We appreciate this suggestion. In accordance with the reviewer's suggestion, we have additionally created various reported CtIP mutants and analysed their functionality in the LoAD system. The results were shown in newly added Supplementary Figure 10 and described in pages 4–5 of the main text. Discussion was also added in page 8.

REVIEWERS' COMMENTS:

Reviewer #1 (Remarks to the Author):

After reading the revised version of the manuscript I now support its publication in Nature Communications. The authors have taken on board all the suggestions by both reviewers and have adequately address them. In my opinion, it can be published as it is but it will be helpful to add a few sentences on the discussion comparing the LoAD method they are presenting with the Cas9-CtIP fusion described by Charpentier et al last march in the same journal. But is just a suggestion only.

Reviewer #2 (Remarks to the Author):

All the comments were well addressed in the revised manuscript.

It is worth to point out that, during the revision process, another complementary study entitled "CtIP fusion to Cas9 enhances transgene integration by homology-dependent repair" was published. Charpentier et al shown that adding N-terminal fragment of CtIP to Cas9 is sufficient to stimulate HDR (twofolds), which is consistent with this paper's conclusion. It would be great to mention Charpentier's work in the discussion.

Response to Reviewer #1

We appreciate the great help from the reviewer for the improvement of our paper. A summary of the revisions made in accordance with the comments from the reviewer, together with our response, is shown below. The valuable suggestions of the reviewer are also listed again for explanation.

Remarks to the Author

After reading the revised version of the manuscript I now support its publication in Nature Communications. The authors have taken on board all the suggestions by both reviewers and have adequately address them. In my opinion, it can be published as it is but it will be helpful to add a few sentences on the discussion comparing the LoAD method they are presenting with the Cas9-CtIP fusion described by Charpentier et al last march in the same journal. But is just a suggestion only.

Response: We appreciate this suggestion. In accordance with the reviewer's suggestion, we have added the corresponding discussion in page 8 of the main text.

Response to Reviewer #2

We appreciate the great help from the reviewer for the improvement of our paper. A summary of the revisions made in accordance with the comments from the reviewer, together with our response, is shown below. The valuable suggestions of the reviewer are also listed again for explanation.

Remarks to the Author

All the comments were well addressed in the revised manuscript.

It is worth to point out that, during the revision process, another complementary study entitled "CtIP fusion to Cas9 enhances transgene integration by homology-dependent repair" was published. Charpentier et al shown that adding N-terminal fragment of CtIP to Cas9 is sufficient to stimulate HDR (twofolds), which is consistent with this paper's conclusion. It would be great to mention Charpentier's work in the discussion.

Response: We appreciate this suggestion. In accordance with the reviewer's suggestion, we have added the corresponding discussion in page 8 of the main text.